# Factors influencing the functionality of medicines and therapeutic committees in public health facilities in Uganda: A longitudinal assessment

Benard Nsubuga[1]*, Anthony Ddamba[1], Harriet Akello[2], David Arinaitwe[1], Phillip Ampaire[1], Moses Kamabare[1]

**1** Client Services Department, National Medical Stores, Entebbe, Uganda, **2** Department of Pharmaceuticals and Natural Medicines Ministry of Health, Kampala, Uganda

* bnsubuga22@gmail.com

## Abstract

World health organization estimates that of all medicines used globally, about 50% are used irrationally. This is mostly attributed to overuse of antibiotics and indiscriminate use of injectables, among other things. Globally, medicines and therapeutic committees (MTCs) play a significant role in combating inappropriate medicines use problems. This study explored the extent of the functionality of MTCs in health facilities in Uganda and the associated factors. This was a longitudinal assessment that utilised a semistructured questionnaire that was administered through face-to-face interviews. Panel data analysis techniques were used to compare results between the two waves. This study utilised the ordered logistic regression model and random effects techniques to determine the factors influencing the functionality of MTCs. Overall, the percentage of health facilities with a functional MTC significantly increased from 8.3% (22/264) in wave one to 20.8% (55/265) in wave two (p-value = 0.000). The percentage of MTCs that are partially functioning increased from 15.2% (40/264) in wave one to 29.1% (77/265) in wave two. The percentage of HFs with MTC structures increased from 2.6% to 42.6% in wave two. Health facilities without an MTC significantly reduced from 72% (190/264) in wave one to 7.6% (20/265) in wave two (p-value = 0.0000). The median number of MTC members significantly increased from 13 in wave one to 15 members in wave two (p-value = 0.0112). Level of care, availability of guidelines, availability of subcommittees, and MTC members being active were significantly associated with the performance of MTCs (p-value<0.05). The study established a significant improvement in the functionality of MTCs in health facilities attributed to interventions from national medical stores, Uganda's ministry of health and implementing partners. The study, however, observed little improvement in the full functionality of MTC subcommittees. Continuous medical education interventions should be strengthened at all levels of care.

**Data availability statement:** All relevant data are within the paper and its Supporting Information files.

**Funding:** The author(s) received no specific funding for this work.

**Competing interests:** The authors have declared that no competing interests exist.

## Background

Medicines use problems have become more prevalent in recent times. World health organization (WHO) estimates that of all medicines used globally, about 50% are used irrationally [1]. This problem is more pronounced in the elderly, mostly attributed to polypharmacy tendencies [2,3]. The problems of irrational drug use are mirrored through poor prescribing tendencies, dispensing, and poor patient adherence to use of prescribed medicines [3]. The consequences of inappropriate medicines use range from misuse of scarce resources to adverse therapies clinical consequences and, any intervention intended to promote the rational use of medicines should be integral to the health care management systems. Successful health care systems across the world primarily depend on a myriad of factors including the attributes of leaders governing such health care systems. Globally, medicines and therapeutic committees (MTCs) play a significant role in ensuring appropriate medicines use. MTCs are the operational arm of health facilities (HFs) geared to improve the efficient management and use of essential medicines [4].

MTCs are governance bodies aimed at ensuring optimal quality of health care in the population through providing stewardship for cost-effective use of medicines, thereby preventing insufficient therapeutic effect, adverse drug reactions, preventable side-effects and interactions from medicines, and increasing resistance of bacterial pathogens to antimicrobial medicines [5,4]. This is achieved through performing functions such as advisory to medical staff, pharmacy, and administration; development of policies and procedures; development of HF formulary list of medicines; development of standard treatment guidelines; assessment of medicines use problems; conducting of effective interventions to improve medicines use problems; and to manage medication errors [4,6]. MTC composition involves a multidisciplinary integration of professionals involved in health care provision to the population. These may include infectious disease specialists, paediatricians, pharmacologists, nurses, pharmacists, representatives of the HF management, public health physician, a general practitioner among others [5].

The presence of MTCs considerably varies between developed and developing economies [7,8]. Their existence in developed countries dates to eight decades in time and their composition and roles nearly exhibit the same picture in many developed countries [9]. However, the prevalence and functionality of MTCs in low- and middle-income countries (LMICs) is ambivalently described whose shortcomings manifest from poor funding, poor communication among members, lack of established standards, overburdened committee personnel, and gilded subcommittees that are often non-operational [10]. In Nigeria, for example, Fadare *et. al.* (2018) in their study to establish the presence and functionality of MTCs in the country found minimal presence of MTCs in many of the surveyed hospitals, often characterized by limited or non-functioning subcommittees and this alludes to many countries including Brazil, Jordan, Sierra-Leone among others [9,11,12].

In Uganda, just like in other LMICs, the absence of functioning MTCs is a common phenomenon [10]. This has resulted into inappropriate medicine use problems in addition to increased instances of antimicrobial resistance (AMR) problems, length of

hospital stays, increased cost of treatment, and drug toxicity and fatalities [13]. The inability by the HFs to develop hospital formularies, polypharmacy, inadequate labelling of medicines in stores, overuse of antibiotics and low adherence to standard treatment guidelines are still being reported in many studies conducted about medicines use in HFs in Uganda [14–17]. The absence of MTC and its subcommittees compromises the quality of medicines prescribing, access and management of medicines within the HF since such committees provide a forum used by health care providers to facilitate knowledge sharing, professional development as well as encouraging mentorship through continuous medical education (CMEs) [10,18,19].

Despite the presence of several policy documents including MTC manuals (2018 & 2025) [20,21], coupled with collaboration among partners in the health sector such as global fund (GF), infectious diseases institute (IDI), and management sciences for health (MSH), the presence of functional MTCs in HFs is still a challenge. To the best of our knowledge, the existence of literature about the status of MTCs in Uganda in recent years is ambivalent. The few studies available largely focused on the alignment of MTC guidelines to existing structure and scope of the committees without necessarily attempting to unearth factors hindering optimal performance of MTCs and the extent to which MTCs function. Moreover, unlike previous research that has often been limited in scope focusing on a small number of hospitals [9,22], this study enhances the generalizability of findings and contributes robust empirical knowledge to inform both policy and practice in Uganda's health care sector.

This study, therefore used panel data techniques on data that was collected from all government-owned health facilities to establish the trend in the functionality of MTCs and its associated factors between two interventional waves.

## Materials and methods

### Study design

This was a longitudinal assessment study, following up on the same panels for which data were collected at baseline. The panel in this case was the health facilities. Therefore, panel data analysis techniques were used to compare results between the two waves. This study was quantitative in nature and utilised quantitative data analysis methods. The panel data in this study contained data points for each HF that was collected at two different points in time.

### Study setting and target population

The public health care landscape in Uganda has national referrals (NRs) at the apex designed to offer highly specialised medical services. This is followed by regional referral hospitals (RRHs) that offer specialised medical healthcare at the regional level. Below RRHs are the district hospitals commonly referred to as general hospitals (GHs), which offer high level primary health care services within districts. Finally, health center IVs (HCIVs) offer primary health care services at county level jurisdictions under a decentralised administrative system [23]. This study was conducted in all public HFs including HC4s, GHs, RRHs, and NRs. The target participants included MTC chairpersons, MTC secretaries, MTC members, hospital directors, medical superintendents, HC4 In-charges, and Dispensers. In most HFs, pharmacists double as MTC secretaries. One response was submitted from each HF.

### Sampling procedure and selection of respondents

The study used a census method that covered all government-owned HFs including all HC4s (194), all GHs (48), all RRHs (17), and all NRs (6). A census approach was preferred for the units of analysis to facilitate presentation of findings and MTC performance levels for all HFs served by government as this would facilitate and trigger policy intervention stages for each HF. The primary respondent in each HF was the pharmacist who doubles as the MTC secretary. However, in the absence of the MTC secretary and MTC chairperson, one MTC member was purposively selected to represent the HF MTC in this study. A 100% (265/265) response rate was achieved.

## Data collection process

The study used a semistructured survey questionnaire to collect data from the study respondents. The semistructured questionnaire was developed by the research team for wave one and was adopted by the same research team in wave two. Validation and pretesting of the tool were carried out in wave one. This was done to achieve internal consistency and to ensure that the survey tool collects data aligned to the study objectives. The face-to-face interview method was used, facilitated by Kobo tool kit, a web-based data collection engine. Kobo tool kit aided real time transmission of data from the field teams.

The study was conducted between January and February 2025. The data collection team comprised of 20 pharmacists who were carefully recruited due to the diverse knowledge they possess in supply chain and medicines management. A one-day training workshop was held to equip them with the required interview skills as well as understanding the objective of this study and how to administer questions in the survey tool. The data collection process was supervised by the principal investigator and the deputy principal investigator.

## Data management and analysis

The data collected were transferred from the kobo toolbox into STATA 18.5 (statacorp Texas, USA). The data cleaning process encompassed data editing and the variable transformation processes including defining dummy variables for categorical regressors. Univariate analysis encompassed generating descriptive summary statistics (mean, median, percentages and frequencies). Bivariate data analysis techniques were utilised to establish the statistically significant relationship between any two variables of interest. This was facilitated by Pearson's chi-square test, a linear-by-linear chi-square test, and Wilcoxon rank-sum test, a distribution free test. The type of chi-square used was dictated by the structure of the two variables under consideration. The statistical significance was established using 5% level of significance.

## Model estimation

The functionality status was constructed on a four-point scale where 1 = lack of MTC, 2 = Yes, MTC available but not functional, 3 = Yes, MTC available but partially functional, and 4 = MTC, fully functional. The response variable "functionality" depicts a natural ranking, conforming to a natural ordering. This was therefore coded as consecutive integers starting at 1–4. Therefore, to determine the factors influencing the performance of MTCs in HFs, a nonlinear ordinal-logistic regression model (ORM) was used. Secondly, the study utilised specific statistical techniques designed to handle panel data; subject specific (SS) and population average (PA) models. In this study the SS model was used as this would properly allow exploring statistical dependence commonly known as "*autocorrelation*" among the repeated responses within subjects that must be accounted for using an appropriate statistical model rather than assuming average dependence among the repeated responses, which is often the case with PA models [24–26]. Second, the SS model allowed use of the random effects (RE) model, thus facilitating the exploration of time invariant predictors.

For purposes of this report, a fully functional MTC was defined as one such committee that held regular meetings, maintained documentation of operations, and performed critical roles such as evaluating medicine use and logistical functions and or developing Institutional EMHS lists.

## Ethics approval and consent to participate

The study was conducted in accordance with the principles of the Helsinki Declaration. All participants provided written informed consent, and the study was approved by the research and ethics committee (REC) of Makerere University School of Health Sciences with reference number MAKSHSREC-2024–768 The study was further approved by the Uganda National Council for Science and Technology, under approval number HS5391ES. Prior to the interview process, each participant signed a consent form, accepting to participate in this study.

   

## Results

This section presents findings from wave two of the medicines and therapeutic committees in public HFs. This was a follow-up study providing the current functionality status compared to the baseline assessment, one year apart between the two waves.

### Background characteristics

The study interviewed a total of 265 MTCs comprising 194 HC4s, 48 GHs, 17 RRHs, and 6 NRs. Overall, teaching HFs accounted for 14.3% (38/265) of the total sample while 85.7% (227/265) were non-teaching HFs. Table 1 shows that the highest percentage of respondents were MTC chairpersons (19.2%, 51/265) followed by assistant inventory management officers (18.1%, 48/265), dispensers (17.7%, 47/265), while pharmacists accounted for 17% (45/265). Furthermore, the highest proportion of respondents (35.5%,94/265) had served for over six years in their current positions while 35.1% (93/265) had served for about three to five years.

### Functionality of MTCs

Fig 1 shows that the percentage of HFs with a functional MTC increased significantly from 8.3% (22/264) in wave one to 20.8% (55/265) in wave two (p-value<0.05). The findings revealed a sharp fall in the proportion of HFs with no MTC in place from 72% (190/265) in wave one to only 7.6% (20/265) in wave two. The biggest improvement in having a fully functional MTC

**Table 1. Summary of background characteristics (n=265).**

| | Nature of HF | | | | | | |
|---|---|---|---|---|---|---|---|
| | Non-Teaching HF | | Teaching HF | | Total | | |
| Level of Care | *Freq.* | % | *Freq.* | % | *Freq.* | % | |
| HC4 | 188 | 96.9 | 6 | 3.1 | 194 | 100 | |
| GH | 36 | 75 | 12 | 25 | 48 | 100 | |
| RRH | 3 | 17.6 | 14 | 82.4 | 17 | 100 | |
| NR | | | 6 | 100 | 6 | 100 | |
| Total | 227 | 85.7 | 38 | 14.3 | 265 | 100 | |
| | Interviewee position in the HF | | | | | | |
| | | HC4 | GH | RRH | NR | Total | |
| Title | | *Freq.* | *Freq.* | *Freq.* | *Freq.* | *Freq.* | % |
| MTC Chairperson | | 44 | 6 | | 1 | 51 | 19.2% |
| AIMO/IMO | | 43 | 5 | | | 48 | 18.1% |
| Dispenser | | 39 | 8 | | | 47 | 17.7% |
| Pharmacist | | 2 | 22 | 16 | 5 | 45 | 17.0% |
| MTC Member | | 30 | 4 | 1 | | 35 | 13.2% |
| Director/Superintendent | | 30 | 3 | | | 33 | 12.5% |
| Other | | 6 | | | | 6 | 2.3% |
| Total | | 194 | 48 | 17 | 6 | 265 | 100.0% |
| | Years Served in the HF | | | | | | |
| | | HC4 | GH | RRH | NR | Total | |
| Years | | *Freq.* | *Freq.* | *Freq.* | *Freq.* | *Freq.* | % |
| 6 years and above | | 58 | 25 | 8 | 3 | 94 | 35.5% |
| 3 to 5 years | | 66 | 16 | 8 | 3 | 93 | 35.1% |
| 1 to 2 years | | 42 | 7 | | | 49 | 18.5% |
| Less than 1 year | | 28 | | 1 | | 29 | 10.9% |
| Total | | 194 | 48 | 17 | 6 | 265 | 100.0% |

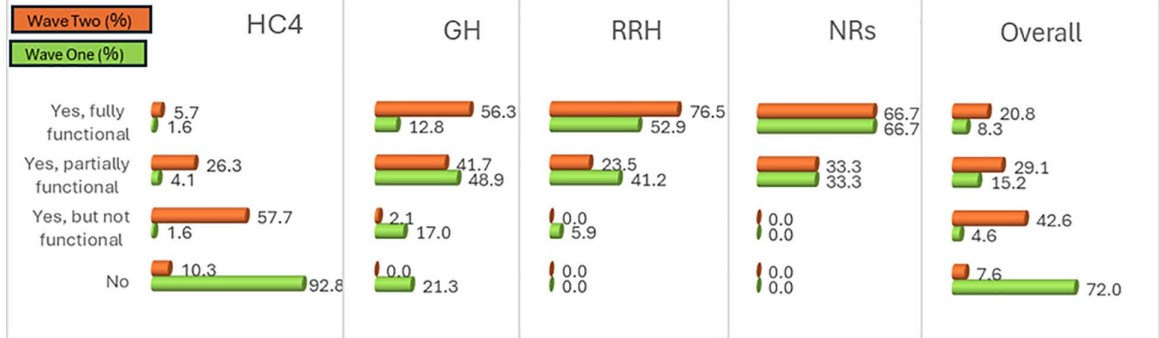

**Fig 1. Functionality Status of MTCs in HFs.**

between the two waves was observed in GHs (wave one = 12.8%, wave two = 56.7%) and RRHs (wave one = 52.9%, wave two = 76.5%). As in the baseline, RRHs still reported the highest percentage of HFs with a fully functional MTC (13/17).

## Testing for the distribution of functionality across levels of care

The study further explored the extent of the linear relationship between the functionality status of MTCs and the HF level of care. A linear-by-linear chi-square test was used to establish whether a linear trend existed between the two variables. Furthermore, a significantly positive relationship was established between the two waves and the functional status (p-value = 0.00 < 0.05), demonstrating improved performance overtime.

Table 2 shows that the mean score on the functional real line at the HC4 level of care in wave two was 2.27 (SD = 0.72) compared to 1.14 (SD = 0.55) in wave one. The mean score at the GH level in wave two was 3.54 (SD = 0.54) compared to 2.53 (SD = 0.97) in wave one. The mean score at the RRH level in wave two was 3.76 (SD = 0.44) compared to 3.47 (SD = 0.62) in wave one. The mean score at the NR level remained constant in both waves (mean = 3.67, SD = 0.52). The

**Table 2. Testing for the distribution of functionality across levels of care.**

| Distribution of functionality across levels of care (wave two) | | | | | | | | | | |
|---|---|---|---|---|---|---|---|---|---|---|
| MTC functionality | HC4 | | GH | | RRH | | NR | | Total | |
| | Freq | % | Freq | % | Freq | % | Freq | % | Freq | % |
| No | 20 | 10.3 | 0 | | 0 | | 0 | | 20 | 7.5 |
| Yes, but not functional | 112 | 57.7 | 1 | 2.1 | 0 | | 0 | | 113 | 42.6 |
| Yes, partially functional | 51 | 26.3 | 20 | 41.7 | 4 | 23.5 | 2 | 33.3 | 77 | 29.1 |
| Yes, fully functional | 11 | 5.7 | 27 | 56.2 | 13 | 76.5 | 4 | 66.7 | 55 | 20.8 |
| Total | 194 | 100 | 48 | 100 | 17 | 100 | 6 | 100 | 265 | 100 |

| Testing for a linear trend in the mean scores on functionality across levels of care in the two waves | | | | |
|---|---|---|---|---|
| | HC4 | GH | RRH | NR | Total |
| Wave one: Mean (SD) | 1.14 (0.55) | 2.53 (0.97) | 3.47 (0.62) | 3.67 (0.52) | 1.6 (1.02) |
| Wave two: Mean (SD) | 2.27 (0.72) | 3.54 (0.54) | 3.76 (0.44) | 3.67 (0.52) | 2.63 (0.9) |
| Linear by Linear Chi-square: by level of care | 9.804 (0.0000*) | | | |
| Linear by Linear Chi-square: by wave | 10.879 (0.0000*) | | | |

Source: Primary data; *Significant at 5%.

functional status score increased with an increase in the level of care, exhibiting a statistically significant positive linear trend (p-value=0.00<0.05). This means that greater impact is currently realised in HFs that offer more specialised services with more specialist cadres who have a high-level understanding of the role of MTCs in stewarding appropriate medicines use and management. Furthermore, a significantly positive relationship was established between the two waves and the functional status (p-value=0.00<0.05), demonstrating improved performance overtime.

## MTC functionality transitional probabilities

Table 3 describes the likelihood with which HF MTCs moved from one category of functionality into another, below or above the category. The rows reflect numbers in wave one while the columns represent final destinations. For example, in wave one, 190 HFs lacked an MTC, however, in wave two, this number reduced significantly to 20 HFs. 55.3% (105/190) of the HFs that lacked an MTC at baseline had established MTCs in place, although they were not functional, 25.3% (48/190) transitioned to partial functionality while 9% (17/190) transitioned to full functionality status. Table 3 below further shows that the probability of transitioning from having a partially functional MTC to a fully functional MTC considering wave one and wave two results was 52.5% higher compared to transitioning from having a nonfunctional MTC to a fully functional MTC (33.3%).

## Reasons for the lack of a functional MTC

Table 4 presents some of the challenges contributing to the suboptimal performance of MTCs in HFs. These are distributed both by wave and level of care. Statistical significance between waves was tested using Pearson's chi-square test at a 95% level of confidence. It was established that lack of adequate knowledge in MTC operations (wave one=89.4%, wave two=73.1%) and lack of financial support (wave one=64.8%, wave two=51.9%) are still prevalent in hampering MTC performance in the HFs. All the challenges saw a downward trend between the two waves and the differences in each of the challenges between the two waves were statistically significant (p-value=0.00<0.05) except the challenge of lack of commitment from MTC members (p-value=0.753>0.05). Overall, the challenges facing the operational status of MTCs in HFs are gradually waning. Table 4 below further shows that these challenges are more pronounced in HC4s, and GHs to a lesser extent, but almost near to no concentration in RRHs and NRs. The zeal and focus to operationalize MTCs should be targeted to HC4s rather than any other level of care, especially leveraging the advantage that the MTC structure is in place for majority of the HC4s and existence of commitment from the MTC members.

## Appointment of MTC members

Fig 2 shows that overall, 86% of MTCs had members officially appointed in wave two compared to only 35.7% in wave one. All levels of care scored above 80% with RRHs having the highest percentage of officially appointed members (wave one=68.8%, wave two=94.1%) followed by GHs (wave one=26.5%, wave two=91.7%). HC4s recorded the biggest improvement between the two waves, moving from no MTCs with officially appointed members in wave one to 83.7% in wave two.

Table 3. MTC functionality transitional probabilities.

| MTC functionality | No | | Yes, but not functional | | Yes, partially functional | | Yes, fully functional | | Wave one | |
|---|---|---|---|---|---|---|---|---|---|---|
| | Freq. | % | Freq. | % | Freq. | % | Freq. | % | Freq. | % |
| No | 20 | 10.5 | 105 | 55.3 | 48 | 25.3 | 17 | 9.0 | **190** | 100.0 |
| Yes, but not functional | 0 | 0.0 | 3 | 25.0 | 5 | 41.7 | 4 | 33.3 | **12** | 100.0 |
| Yes, partially functional | 0 | 0.0 | 4 | 10.0 | 15 | 37.5 | 21 | 52.5 | **40** | 100.0 |
| Yes, fully functional | 0 | 0.0 | 1 | 4.6 | 8 | 36.4 | 13 | 59.1 | **22** | 100.0 |
| Wave two | **20** | **7.6** | **113** | **42.8** | **76** | **28.8** | **55** | **20.8** | **264** | **100.0** |

**Table 4. Distribution of challenges for lack of a functional MTC by wave and level of care.**

| Distribution of challenges by wave (%) | | | | |
|---|---|---|---|---|
| Challenge | Wave one | Wave two | P-value | Conclusion |
| Knowledge gap in MTC operations | 89.4 | 73.1 | *0.000** | *Statistically significant* |
| lack of financial support | 64.8 | 51.9 | *0.006** | *Statistically significant* |
| Lack of commitment from members | 29.2 | 27.9 | *0.753* | *Not statistically significant* |
| Lack of will and support from hospital administration | 8.1 | 15.4 | *0.016** | *Statistically significant* |
| Other specify | 14.4 | 42.8 | *0.000** | *Statistically significant* |
| Overall p-value | | | *0.000** | *Overall, statistically significant* |

| Distribution of challenges by level of care (%) | | | | | | | | |
|---|---|---|---|---|---|---|---|---|
| | HC4 | | GH | | RRH | | NR | |
| | Wave one | Wave two | Wave one | Wave two | Wave one | Wave two | Wave one | Wave two |
| Knowledge gap in MTC operations | 77.5 | 63.0 | 10.6 | 8.7 | 0.9 | 1.0 | 0.4 | 0.5 |
| lack of financial support | 49.2 | 43.3 | 11.9 | 6.7 | 3.0 | 1.0 | 0.9 | 1.0 |
| Lack of commitment from members | 18.6 | 20.7 | 8.5 | 5.3 | 2.1 | 1.4 | 0.0 | 0.5 |
| Lack of will and support from hospital administration | 3.0 | 11.5 | 4.2 | 3.4 | 0.4 | 0.5 | 0.4 | 0.0 |
| Other specify | 12.3 | 38.9 | 1.7 | 3.4 | 0.0 | 0.5 | 0.4 | 0.0 |

Source: Primary Data; *Significant at 5%.

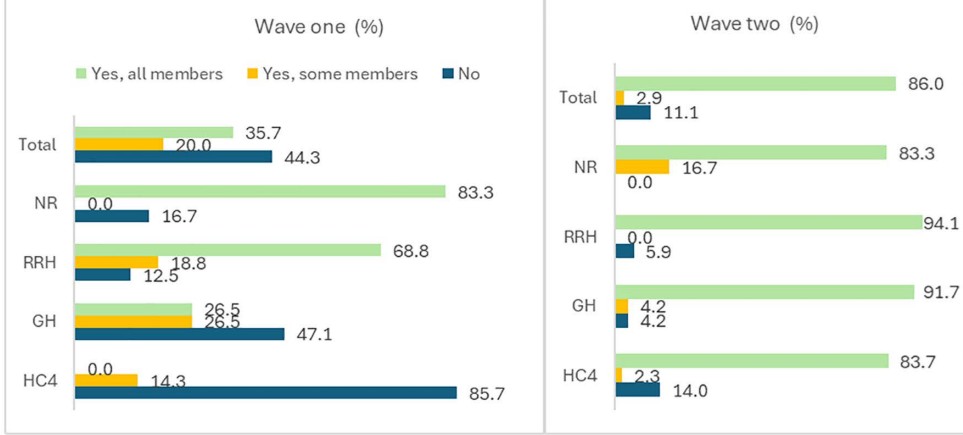

**Fig 2. Appointment of MTC members (n = 265).**

## Average number of MTC Members

The median number of MTC members was highest at the RRH level in both waves and the gaps are so wide compared to other HF levels of care. Overall, the median number of participants was 13 (min = 0, max = 32) in wave one compared to 15 (min = 5, max = 30) in wave two. The study went further to statistically establish whether the median numbers significantly varied between the two waves. Wilcoxon rank-sum test, a distribution free test was used to test for the equality of group medians using ranks. Results in Table 5 revealed that the median number of MTC members between the two waves was significantly different from each other (p-value = 0.0112 < 0.05).

**Table 5.  Median number of MTC members by level of care.**

| Level of care | Wave One | | | Wave Two | | |
|---|---|---|---|---|---|---|
| | Median | Min | Max | Median | Min | Max |
| HC4 | 9.5 | 0 | 30 | 14 | 5 | 22 |
| GH | 12 | 5 | 23 | 16 | 11 | 27 |
| RRH | 18 | 15 | 32 | 19 | 15 | 30 |
| NR | 12.5 | 10 | 32 | 15 | 13 | 22 |
| Total | 13 | 0 | 32 | 15 | 5 | 30 |
| Wilcoxon rank-sum test | | | | 0.0112* | | |

Source: Primary data; Significant at 5%.

## Availability of policies and procedures

Fig 3 shows the trend in the availability of policies and procedures in HFs between the two waves. The ToRs and MTC guidelines recorded the highest percentage change in their availability in HFs between the two waves (wave one = 44.6%, wave two = 88.2%) and (wave one = 66.2%, wave two = 82.5%) respectively, while PV (wave one = 46.0%, wave two = 27.8%) and AMS (wave one = 24.3%, wave two = 13.9%) guidelines registered a significant decline. STG and EMHS list maintained a high presence between the two waves.

The study further explored the statistical significance of the changes in the availability of policies and procedures between wave one and wave two. Pearson's chi-square test was used. Overall, the changes in the availability of policies and procedures between wave one and wave two were statistically significant (p-value = 0.000 < 0.05). This means that as we move from the initial stages of the intervention into the operational phase, availing policy documents to HF MTCs to guide their operations is paramount. (Table 6)

## Availability of MTC subcommittees

Table 7 shows that the supply chain subcommittee is the most prevalent, although its full functionality status declined from 37.0% in wave one to 25.3% in wave two. Besides the research subcommittee, all other subcommittees are in place regardless of operational status, unlike in wave one where lack of subcommittees in place was more prevalent. The highest presence and improvement were observed at the RRH level, just like in wave one.

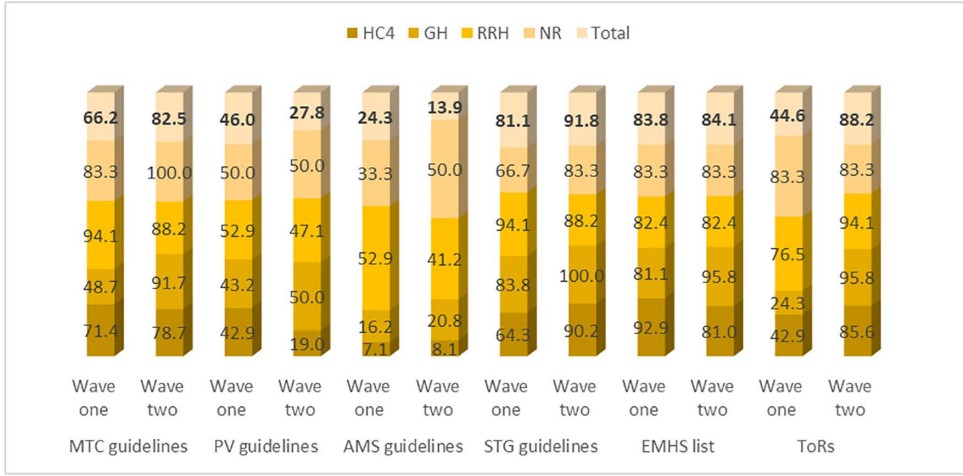

**Fig 3.  Percentage availability of policies and procedures.**

**Table 6. Availability of policies and procedures.**

| Guidelines | Cases (Frequency) | | |
|---|---|---|---|
| | **Wave one** | **Wave two** | **Group p-value (unadjusted)** |
| MTC guidelines | 49 | 202 | 0.003* |
| PV guidelines | 34 | 68 | 0.003* |
| AMS guidelines | 18 | 34 | 0.031* |
| STG guidelines | 60 | 225 | 0.009* |
| EMHS list | 62 | 206 | 0.974 |
| ToRs | 33 | 216 | 0.000* |
| Cases: Overall P-value (adjusted) | 73 | 243 | 0.000* |

Source Primary data; *significant at 5%.

**Table 7. Availability of MTC subcommittees (wave two, n=265).**

| | Availability of PV subcommittee | | | | | | | | | |
|---|---|---|---|---|---|---|---|---|---|---|
| | **HC4** | | **GH** | | **RRH** | | **NR** | | **Total** | |
| **Functionality status** | **Wave one** | **Wave two** | **Wave one** | **Wave two** | **Wave one** | **Wave two** | **Wave one** | **Wave two** | **Wave one** | **Wave two** |
| Very Functional | 21.4 | 1.2 | 16.2 | 35.4 | 29.4 | 58.8 | 16.7 | 16.7 | 20.3 | 12.3 |
| Partially functional | 14.3 | 16.8 | 24.3 | 41.7 | 41.2 | 23.5 | 50.0 | 50.0 | 28.4 | 23.0 |
| Available but not functional | 14.3 | 74.0 | 5.4 | 18.8 | 29.4 | 17.7 | 33.3 | 16.7 | 14.9 | 57.8 |
| Not in place | 50.0 | 8.1 | 54.1 | 4.2 | 0.0 | 0.0 | 0.0 | 16.7 | 36.5 | 7.0 |
| Total | 100.0 | 100.0 | 100.0 | 100.0 | 100.0 | 100.0 | 100.0 | 100.0 | 100.0 | 100.0 |
| | **Availability of AMS subcommittee** | | | | | | | | | |
| | **HC4** | | **GH** | | **RRH** | | **NR** | | **Total** | |
| **Functionality status** | **Wave one** | **Wave two** | **Wave one** | **Wave two** | **Wave one** | **Wave two** | **Wave one** | **Wave two** | **Wave one** | **Wave two** |
| Very Functional | 7.1 | 2.3 | 2.9 | 20.8 | 17.7 | 52.9 | 33.3 | 33.3 | 9.7 | 10.3 |
| Partially functional | 21.4 | 11.0 | 8.6 | 45.8 | 58.8 | 35.3 | 16.7 | 50.0 | 23.6 | 20.5 |
| Available but not functional | 21.4 | 78.0 | 8.6 | 29.2 | 23.5 | 11.8 | 50.0 | 16.7 | 18.1 | 62.3 |
| Not in place | 50.0 | 8.7 | 80.0 | 4.2 | 0.0 | 0.0 | 0.0 | 0.0 | 48.6 | 7.0 |
| Total | 100.0 | 100.0 | 100.0 | 100.0 | 100.0 | 100.0 | 100.0 | 100.0 | 100.0 | 100.0 |
| | **Availability of supply chain subcommittee** | | | | | | | | | |
| | **HC4** | | **GH** | | **RRH** | | **NR** | | **Total** | |
| **Functionality status** | **Wave one** | **Wave two** | **Wave one** | **Wave two** | **Wave one** | **Wave two** | **Wave one** | **Wave two** | **Wave one** | **Wave two** |
| Very Functional | 21.4 | 8.6 | 37.8 | 66.7 | 56.3 | 70.6 | 16.7 | 50.0 | 37.0 | 25.3 |
| Partially functional | 35.7 | 19.0 | 21.6 | 16.7 | 18.8 | 17.7 | 50.0 | 33.3 | 26.0 | 18.8 |
| Available but not functional | 21.4 | 64.9 | 8.1 | 10.4 | 18.8 | 11.8 | 33.3 | 16.7 | 15.1 | 49.4 |
| Not in place | 21.4 | 7.5 | 32.4 | 6.3 | 6.3 | 0.0 | 0.0 | 0.0 | 21.9 | 6.5 |
| Total | 100.0 | 100.0 | 100.0 | 100.0 | 100.0 | 100.0 | 100.0 | 100.0 | 100.0 | 100.0 |
| | **Availability of Research subcommittee** | | | | | | | | | |
| | **HC4** | | **GH** | | **RRH** | | **NR** | | **Total** | |
| **Functionality status** | **Wave one** | **Wave two** | **Wave one** | **Wave two** | **Wave one** | **Wave two** | **Wave one** | **Wave two** | **Wave one** | **Wave two** |
| Very Functional | 0.0 | 0.0 | 2.8 | 0.0 | 6.3 | 12.5 | 16.7 | 20.0 | 4.3 | 1.3 |
| Partially functional | 0.0 | 0.0 | 5.6 | 2.2 | 25.0 | 12.5 | 33.3 | 20.0 | 11.4 | 1.7 |
| Available but not functional | 16.7 | 14.6 | 5.6 | 6.7 | 12.5 | 31.3 | 0.0 | 0.0 | 8.6 | 13.9 |
| Not in place | 83.3 | 85.4 | 86.1 | 91.1 | 56.3 | 43.8 | 50.0 | 60.0 | 75.7 | 83.1 |
| Total | 100.0 | 100.0 | 100.0 | 100.0 | 100.0 | 100.0 | 100.0 | 100.0 | 100.0 | 100.0 |

Source: Primary data.

## Planning and budgeting

The findings show an upward trend in the percentage of MTCs with a workplan, and budget included in the HF workplan and budget for the MTC work (wave one=16.2%, wave two=27.1%). The availability of a workplan and a budget for the MTC work is significantly associated with an increase in wave number (p-value=0.000<0.05). Any further interventions as time goes by are likely to positively influence the formulation of MTC workplans and budgeting to smoothly run the MTC work. Table 8 shows that GHs (wave one=8.1%, wave two=43.8%), RRHs (wave one=23.5%, wave two=41.2%), and NRs (wave one=16.7%, wave two=33.3%) had the highest percentage of MTCs whose workplan is incorporated into the HF workplan compared to their counterparts at HF level (wave one=28.6%, wave two=20.8%) that recorded a shortfall of nearly 8 percentage points.

## MTC meetings

Fig 4 shows that among the NRs, in wave two, 66.7% (4/6) conducted MTC meetings quarterly while Butabika NRH and Mulago NRH held MTC meetings bimonthly and monthly respectively. Among the RRHs, the proportion of MTC meetings held bimonthly as stipulated in the MTC manual increased from 50% (8/16) to 58.8% (10/17) in wave two while those holding meetings quarterly increased from 12.5% (2/16) in wave one to 35.3% (6/17) in wave two. Improvement

**Table 8. Availability of budget and workplan (n; wave one=74, wave two, n=244).**

| Level of Care | MTC has a work plan and a budget | | MTC has a work plan and a budget but it is not included in the HF work plan and budget | | MTC has a work plan but without a budget | | No | |
|---|---|---|---|---|---|---|---|---|
| | Wave one | Wave two | Wave one | Wave two | Wave one | Wave two | Wave one | Wave two |
| HC4 | 28.6 | 20.8 | | 5.8 | 7.1 | 8.7 | 64.3 | 64.7 |
| GH | 8.1 | 43.8 | | 25.0 | 24.3 | 8.3 | 67.6 | 22.9 |
| RRH | 23.5 | 41.2 | | 17.7 | 47.1 | 23.5 | 29.4 | 17.7 |
| NR | 16.7 | 33.3 | | 50.0 | 50.0 | 0.0 | 33.3 | 16.7 |
| Total | 16.2 | 27.1 | | 11.5 | 28.4 | 9.4 | 55.4 | 52.1 |
| Waves: Chi-square (p-value) | 26.07 (0.0000*) | | | | | | | |

Source: Primary data; *significant at 5%.

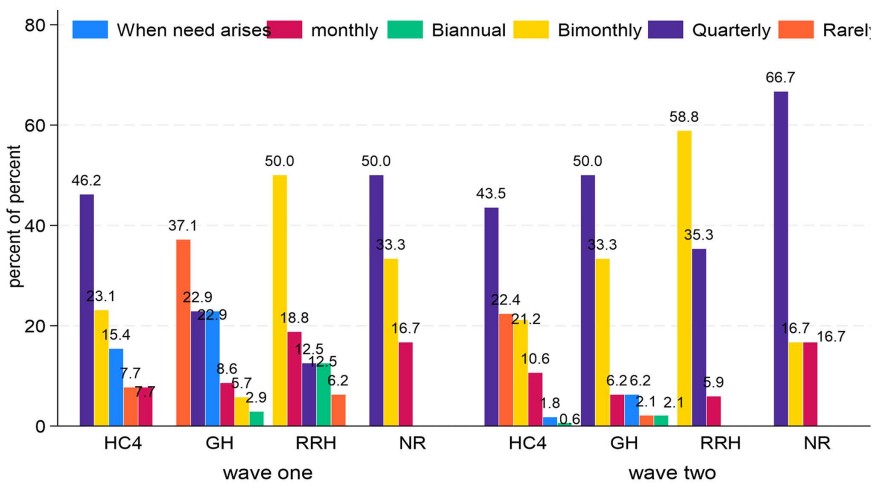

**Fig 4. Frequency of MTC meetings in HFs.**

was observed in GHs where most MTCs held meetings (50%, 24/48) quarterly and 33.3% (16/48) held MTC meetings bimonthly in wave two compared to 22.9% (8/35) and 5.7% (2/35) in wave one respectively. Re-enforcement is needed at the HC4 level. Additionally, the percentage of MTCs documenting their meeting minutes reduced from 77.0% (57/74) in wave one to 61.5% (150/244) in wave two.

The study further explored the significance of the relationship between the frequency of MTC meetings and the level of care. The frequency was numbered on a linear scale as 1= when need arises, 2 = rarely, 3 = biannual, 4 = quarterly, 5 = bimonthly, and 6 = monthly. A linear-by-linear chi-square test, a measure of trend among ordered variables, was used. The findings in Table 9 demonstrate a statistically significant association between the frequency of MTC meetings and the HF level of care (p-value = 0.024 < 0.05). This means that currently, MTCs in NRs, RRHs, and GHs in that order, are more likely to hold MTC meetings regularly compared to their counterparts at the HC4 level.

## MTC interventions through CMEs

Table 10 shows a declining trend in the percentage of HF MTCs conducting CME interventions. At the HF level, NRs (wave one = 33.3, wave two = 83.3%), RRHs (wave one = 70.6, wave two = 82.4%), and GHs (wave one = 44.4%, wave

**Table 9. Test for linear trend between level of care and frequency of MTC meetings.**

| Test for linear trend between level of care and frequency of MTC meetings | | | | |
|---|---|---|---|---|
| Level of care | Wave one | | Wave two | |
| | Freq | Mean Score (SD) | Freq | Mean Score (SD) |
| HC4 | 13 | 3.77 (1.54) | 170 | 3.92 (1.3) |
| GH | 35 | 2.77 (1.59) | 48 | 4.21 (1.11) |
| RRH | 16 | 4.63 (1.15) | 17 | 4.71 (0.59) |
| NR | 6 | 4.67 (0.82) | 6 | 4.5 (0.84) |
| Total | 70 | 3.54 (1.64) | 241 | 4.05 (1.24) |
| Linear by Linear Z-score (P-value) | | | 2.257 (0.0240*) | |

Source: Primary data; *significant at 5%.

**Table 10. MTC interventions through CMEs.**

| MTC Interventions | HC4 (%) | | GH (%) | | RRH (%) | | NR (%) | | Total (%) | |
|---|---|---|---|---|---|---|---|---|---|---|
| | Wave one | Wave two | Wave one | Wave two | Wave one | Wave two | Wave one | Wave two | Wave one | Wave two |
| Advise medical, pharmacy and administrative staff on appropriate medicine use | 42.9 | 21.6 | 44.4 | 70.8 | 70.6 | 82.4 | 33.3 | 83.3 | 49.3 | 37.2 |
| Restrictions (withdrawal of injectables from OPD) | 28.6 | 12.3 | 30.6 | 50.0 | 47.1 | 76.5 | 50.0 | 33.3 | 35.6 | 24.8 |
| Conduct pharmacovigilance activities | 21.4 | 7.6 | 38.9 | 60.4 | 70.6 | 82.4 | 50.0 | 66.7 | 43.8 | 24.8 |
| Monitor the use of standard treatment guidelines | 35.7 | 12.9 | 36.1 | 45.8 | 52.9 | 76.5 | 0.0 | 33.3 | 37.0 | 24.4 |
| Identify medicines use problems | 35.7 | 18.1 | 36.1 | 56.3 | 52.9 | 82.4 | 66.7 | 66.7 | 42.5 | 31.4 |
| Design and implement antimicrobial stewardship activities | 0.0 | 4.1 | 16.7 | 27.1 | 41.2 | 76.5 | 33.3 | 50.0 | 20.6 | 14.9 |
| Conduct appropriate research on medicine use | 0.0 | 0.6 | 5.6 | 8.3 | 23.5 | 58.8 | 0.0 | 33.3 | 8.2 | 7.0 |
| None | 50.0 | 69.0 | 44.4 | 16.7 | 5.9 | 5.9 | 33.3 | 33.3 | 35.6 | 53.3 |
| Others specify | 0.0 | 5.9 | 0.0 | 6.3 | 0.0 | 0.0 | 0.0 | 0.0 | 0.0 | 5.4 |

two = 70.8%), had the highest percentage of MTCs advising health workforce and administration on appropriate medicines use. The percentage of MTCs at HC4 level doing this has reduced from 42.9% in wave one to only 21.6% in wave two.

## Factors influencing the performance of MTCS in HFs

The study conducted inferential analysis to establish the factors affecting the functionality of MTCs in HFs in Uganda. Functionality status was the outcome variable in the analysis below. The overall model was significant at the 95% level of confidence (p-value = 0.000 < 0.05). This means that all the independent variables jointly significantly influence the MTC functionality. The random effects model did not demonstrate high variability between waves (p-value = 0.3158 > 0.05) but was still preferred over the ordered logit model due to the need to account for individual level effects.

Table 11 shows the estimated net effect and the magnitude of the predictor variables on the functionality status of MTCs. Level of care (p-value<0.05), availability of guidelines (p-value<0.05), availability of subcommittees (p-value<0.05), and MTC members being active (p-value<0.05), where identified as the variables significantly affecting MTC performance. NRs (OR=43.2, p-value = 0.002) GHs (OR=9.6, p-value = 0.000), and RRHs (OR=8.9, p-value = 0.018) had higher odds of having functional MTCs compared to HC4s. Additionally, having an AMS subcommittee in a HF is 11 times (p-value = 0.000) more associated with having a functional MTC than with a partially functional MTC and below. HFs with a functional supply chain subcommittee were nearly seven times (OR=6.5, p-value = 0.001) more likely to have a functional MTC than those with a partially functional MTC and below. MTCs with all members active were sixteen times more (p-value = 0.003) likely to have their MTCs fully functional than having partially functional MTC and below compared to those whose members are sometimes active (OR=9.6, p-value = 0.012) and those with no active members. Some variables produced wide confidence intervals (CI) implying less precision in such variables, specifically attributed to higher variability in the data. Usually, very wide confidence intervals are a result of a small sample size; however, in this study, we believe that a sample size of 265 panels was adequate. Estimation methods are usually another common point of encumbrance; however, the nature of our data dictated the estimation method, and we believe that with all other possible methods tested, using ORM and random effects was the best approach.

Table 12 presents the predicted chance of having functional MTCs in HFs. The interventions from NMS, MoH, and implementing partners increased the probability of at least having an MTC in place from 22% in wave one to 45% in wave two. This is attributed to the significant number of HC4 MTCs that were established but not yet operational. Overall, the probability of having fully functional MTCs is still low at 25%. This is attributed to the poor performance at HC4 level, otherwise, NRs (73%), RRHs (66%) and GHs (46%) have high chances of having fully functional MTCs.

## Discussion

This study provides the current state of the functionality of MTCs in HFs in Uganda with comprehensive insights into the factors influencing the functionality of MTCs in addition to evaluating the MTC structure, their operations and the MTC interventions.

### The functionality of MTCs In Uganda

The study found a significant improvement in the prevalence and functionality of MTCs between the two waves under review, moving from 8.3% at the inception of the revitalisation program in 2023 to 20.8% after one year of intervention in 2024. WHO recognizes the establishment of MTCs as one of the strategies to promoting rational medicines use in the HFs. Medicines use problems are a global challenge that cause ineffective treatment, adverse drug events as well as waste of resources. Functional MTCs are envisaged to combat inappropriate prescribing challenges, promote good dispensing practices to avoid medication errors, minimise patients' lack of knowledge about dosing schedules, and provide

**Table 11. Determinants of MTC functionality in public HFs.**

| Functionality status | Adjusted Odds Ratio (95% CI) | P-value |
|---|---|---|
| *Level of care* | | |
| HC4¨ | 1 | |
| GH | 9.6 [2.745, 33.58] | 0.000* |
| RRH | 8.9 [1.461, 54.081] | 0.018* |
| NR | 43.2 [3.817, 488.288] | 0.002* |
| *Nature of HF* | | |
| None teaching HF¨ | 1 | |
| Teaching HF | 2 [0.618, 6.71] | 0.243 |
| *Availability of guidelines* | | |
| MTC guidelines (Yes) | 1 [0.404, 2.282] | 0.927 |
| MTC guidelines (No) ¨ | 1 | |
| PV guidelines (Yes) | 2 [0.925, 4.513] | 0.077** |
| PV guidelines (No) ¨ | 1 | |
| AMS guidelines (Yes) | 3.1 [1.012, 9.5] | 0.048* |
| AMS guidelines (No) ¨ | 1 | |
| National STG (Yes) | 0.6 [0.212, 1.842] | 0.394 |
| National STG (No) ¨ | 1 | |
| EMHS list (Yes) | 1.8 [0.658, 4.712] | 0.26 |
| EMHS list (No) ¨ | 1 | |
| ToR (Yes) | 0.4 [0.112, 1.167] | *0.089** |
| ToR (No) ¨ | 1 | |
| *Subcommittees* | | |
| PV subcommittee (Yes) | 2.2 [0.722, 6.841] | 0.164 |
| PV subcommittee (No) ¨ | 1 | |
| AMS subcommittee (Yes) | 11 [3.137, 38.62] | 0.000* |
| AMS subcommittee (No) ¨ | 1 | |
| Supply chain subcommittee (Yes) | 6.5 [2.142, 19.632] | 0.001* |
| Supply chain subcommittee (No) ¨ | 1 | |
| Research subcommittee (Yes) | 0.1 [0.015, 0.58] | 0.011* |
| Research subcommittee (No) ¨ | 1 | |
| *Number of MTC members* | 1 [0.935, 1.103] | *0.712* |
| *Whether MTC members are active* | | |
| No¨ | 1 | |
| Yes, some members | 9.6 [1.637, 55.75] | 0.012* |
| Yes, all members | 16.3 [2.531, 105.44] | 0.003* |
| *MTC members officially appointed* | | |
| No¨ | 1 | |
| Yes, some members | 0.7 [0.177, 3.116] | 0.683 |
| Yes, all members | 2.2 [0.743, 6.714] | 0.152 |
| *Workplan (Yes)* | 1.5 [0.756, 3.036] | *0.242* |
| *Workplan (No) ¨* | 1 | |

* Significant at 5%, ** Significant at 10%; CI=95% Confidence Interval; ¨=Base category.

**Table 12. Predicted probabilities of MTC functionality.**

| Probability of MTC functionality by Wave | | | | | |
|---|---|---|---|---|---|
| | **Wave Number** | | | | |
| | **Wave One** | **Wave Two** | **Total** | | |
| Pr (MTC available but not functional) | 0.218 | 0.446 | 0.399 | | |
| Pr (MTC, partially functional) | 0.492 | 0.321 | 0.356 | | |
| Pr (MTC, fully functional) | 0.29 | 0.234 | 0.245 | | |
| Probability of MTC functionality by level of care | | | | | |
| | HC4 | GH | RRH | NR | Total |
| Pr (MTC available but not functional) | 0.600 | 0.115 | 0.049 | 0.009 | 0.399 |
| Pr (MTC, partially functional) | 0.344 | 0.422 | 0.296 | 0.259 | 0.356 |
| Pr (MTC, fully functional) | 0.057 | 0.462 | 0.655 | 0.732 | 0.245 |

guidance to patients not adhering to dosing schedules and treatment [4,22,20,21]. The outcome of this is to ensure that the principles of drug acquisition, management and use are achieved.

Whereas significant improvement was observed, 20.8% is still below average. Suboptimal prevalence and functioning of MTCs in our study is consistent with Kimbowa *et. al.,* [22] who found 31.3% (21/67) presence of MTCs in the hospitals in Uganda [22]. However, Kimbowa *et. al.,* [22] did not explore the opportunity to investigate the level of MTC functionality at the various healthcare levels below the RRH level and the trend overtime since they focused on tertiary and RRHs only. The deficiency in the prevalence of fully functional MTCs in Uganda is likened to many other LMICs including Rwanda, Ethiopia, Siera Leone, and Nigeria [17,27]. The deficient functioning of MTCs in HFs may result into uncoordinated medicines and supplies procurements, unnecessary longer hospitalisation for patients, lack of a well-structured support for tackling inappropriate prescribing errors, and failure to cartel drug resistance a first growing health care concern in recent times. Therefore, effective MTCs are desired to manage efficient medicines procurement and medicines use problems in HFs [21,27]. While the interventions have fruition in establishing MTC structures, full functionality is still inadequate especially in HCIVs and should be fast tracked. There is need to address knowledge gap among MTC members and provide sufficient funding towards MTC activities.

## MTC member participation

The WHO recommends that an MTC should be diverse in nature, comprising of professionals from all disciplines. This offers the necessary technical competencies in discussing drug use issues and offers an opportunity to educate members in areas that are not in line with their expertise [4]. Additionally, the MoH MTC manual recommends that an MTC should have between 12–15 MTC members [28]. The median number of MTC members in wave one and wave two was 13 and 15 respectively, demonstrating strong adherence to the recommended number in the MoH MTC manual. The median number in the study findings differed considerably from what Kimbowa *et al.*[22] found in their study. Kimbowa *et al.* [22], This indicated a 10 MTC membership. Kiruyi *et al.* [9] in a systematic review of literature on the MTC structure and roles in LMIC found a median number ranging between 8–25. There is a lack of an internationally recommended number of MTC members other than encouraging such committees to be multidisciplinary [9], even in developed economies the median number of MTC members range from as low as 2 members whereas others boast up-to 40 members [29]. It's worth noting that the findings from this study are more generalizable since it covered a full sample of all public high-level HFs. This is an indication that while establishing the MTC structures during reactivation phase, the level of participation from the health workforce in the HFs has improved. Additionally, MTCs with all members active are more likely to be functional compared to their counterparts. This therefore means that while the composition of MTC members looks sufficient, their regular

availability and commitment is more important to accelerate MTC interventions. This may require concerted efforts including funding for MTC activities from the HFs and other partners as well as motivation for the MTC members.

## The status of MTC subcommittees

The presence of MTC subcommittees is critical if the MTCs are to operate optimally. Subcommittees many a times perform the cardinal role of the MTC. This gives the MTC a leverage to perform its oversight role. The subcommittees are expected to hold more frequent meetings than the main MTC. In this study, the availability of subcommittees significantly improved from 64.5% in wave one to 93% in wave two. These were at varying levels of functionality, with NRs and RRHs more likely to have functional subcommittees compared GHs and HC4s. The most prevalent and functional subcommittees were the supply chain subcommittee and the AMS subcommittee. The supply chain subcommittee is mandated to spearhead inventory management practices in the HF while the AMS subcommittee is charged with the responsibility of assisting the main MTC in the management of antimicrobials [28]. In many countries, the AMS and the supply chain subcommittee are the most prevalent, although the infection prevention and control (IPC) is also more common [9,30,31]. For many years, the most vibrant subcommittee in Uganda has been the supply chain subcommittee. This is due to the need to ensure appropriate medicines selection, quantification and management as aligned in the national medicines policy 2015 [22]. In recent times, the AMS subcommittee has gained more attention in Uganda due to the interventions from partners like the MSH program under MTaPS whose primary goal was to control antimicrobial resistance [1]. The emerging trends in antimicrobial resistance in medicines use require functional AMS subcommittees to steward the correct use of medicines. This committee can set medicines use agenda, develop medicines use policies and procedures, and drive the collaboration drive among all healthcare professionals in a HF. Absence of such a committee imply slowness in advancing correct medicines use strategies. The findings in this study are consistent with Kimbowa *et. al.* (2024), who found high presence of AMS and supply chain subcommittees and a moderate presence of the PV and other subcommittees in hospital MTCs in Uganda [22]. In Other LMICs like Nigeria and Sierra Leone, the presence of the AMS subcommittee was adequate in Sierra Leone, but was lacking in many Nigerian hospitals, where, twenty five percent of hospitals had this committee and the most prevalent subcommittee was instead the IPC subcommittee, found in 75% of the hospitals [12,31]. The study in Sierra Leone was however conducted on a sample of seven hospitals, and their findings may lack generalisability. It was however well grounded with mixed method approach that combined quantitative and qualitative insights, unlike our study that used only quantitative insights.

## Medicines use interventions

In HFs, MTCs are required to spearhead the promotion of rational medicine use [21]. CMEs are among the activities MTCs should prioritize for implementation. The study found significant progress in RRHs, NRs, and GHs, however, minimal progress in ensuring that CME activities are conducted was observed in HC4s. Antimicrobial Resistance (AMR) remains a global challenge, posing threats to human health and existence [32]. In Uganda, this problem is accelerated by lack of knowledge, overuse, misuse, and abuse of antimicrobials such as non-adherence to dosage, self-medication, and purchase of drugs without prescription [33]. Therefore, the need to design and implement interventions that provide stewardship towards regulating and controlling the use of antimicrobials to curb AMR is one of the drivers for advancing health care knowledge to protect the current and future generations [32].

## Conclusions

The study established a significant improvement in the presence and functionality level of MTCs in public HFs. Whereas the prevalence of fully functional MTCs is still suboptimal, the presence of MTCs in HFs has significantly increased between the two waves. While the presence of subcommittees is commendable, their functionality level needs improvement by transitioning from partial to full functionality. Capacity building strategies should be drawn and implemented to

equip MTC members with knowledge relevant to MTC operations. Finally, CME interventions should be strengthened at all levels of care.

## Supporting information

**S1 Table. Raw data.**
(XLSX)

**S1 File. Data Collection Tool.**
(PDF)

## Acknowledgments

NMS would like to express sincere thanks to all who participated in the data collection exercise. Special thanks go to the Ministry of Health (MoH) and the Department of Pharmaceuticals and Natural Medicines (DPNM) for providing guidance. Hospital directors, medical superintendents, pharmacists, HC4 In-charges, MTC chairpersons, MTC secretaries, and MTC members in public hospitals who provided valuable information during the interviews. We also wish to extend our sincere gratitude to NMS pharmacists in the regions who provided guidance to all stakeholders that were interviewed.

## Author contributions

**Conceptualization:** Benard Nsubuga, Anthony Ddamba, Harriet Akello, David Arinaitwe, Phillip Ampaire, Moses Kamabare.

**Data curation:** Benard Nsubuga, David Arinaitwe.

**Formal analysis:** Benard Nsubuga.

**Methodology:** Benard Nsubuga, David Arinaitwe, Phillip Ampaire.

**Resources:** Moses Kamabare.

**Supervision:** Benard Nsubuga, Anthony Ddamba, Harriet Akello, David Arinaitwe, Phillip Ampaire, Moses Kamabare.

**Validation:** Benard Nsubuga, Anthony Ddamba, David Arinaitwe, Phillip Ampaire.

**Visualization:** Benard Nsubuga.

**Writing – original draft:** Benard Nsubuga, Anthony Ddamba, Harriet Akello, David Arinaitwe, Phillip Ampaire.

**Writing – review & editing:** Benard Nsubuga, Anthony Ddamba, Harriet Akello, David Arinaitwe, Phillip Ampaire.

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
