## [Decision Letter · Decision Letter 0]

12 Sep 2025

Dear Dr. Nsubuga,

Thank you for submitting your manuscript to PLOS ONE. After careful consideration, we feel that it has merit but does not fully meet PLOS ONE’s publication criteria as it currently stands. Therefore, we invite you to submit a revised version of the manuscript that addresses the points raised during the review process.

We look forward to receiving your revised manuscript.

Kind regards,

Joseph Olusesan Fadare

Academic Editor

PLOS ONE

Journal Requirements:

2. We note that your Data Availability Statement is currently as follows: All relevant data are within the manuscript and in Supporting Information files.

5. We note you have included a table to which you do not refer in the text of your manuscript. Please ensure that you refer to Table 1,16 and 19 in your text; if accepted, production will need this reference to link the reader to the Table.

6. Please include captions for your Supporting Information files at the end of your manuscript, and update any in-text citations to match accordingly. Please see our Supporting Information guidelines for more information: http://journals.plos.org/plosone/s/supporting-information

Reviewers' comments:

Reviewer's Responses to Questions

**Comments to the Author**

1. Is the manuscript technically sound, and do the data support the conclusions?

Reviewer #1: No

Reviewer #2: Partly

2. Has the statistical analysis been performed appropriately and rigorously?

Reviewer #1: Yes

Reviewer #2: Yes

3. Have the authors made all data underlying the findings in their manuscript fully available?

Reviewer #1: Yes

Reviewer #2: Yes

4. Is the manuscript presented in an intelligible fashion and written in standard English?

Reviewer #1: No

Reviewer #2: Yes

Reviewer #1: General

I get the impression that this is a master thesis that someone wants to publish as a whole? With all the text, very few researchers will read it, and it is difficult to discern the important parts of methodology and results when it is drowned in so much text.

The main body of text is 45 pages (excluding references, front page and figures). I know there is no limit to texts for PLOS One, but I think this is much too long for the findings in this study. The background could easily by reduced to 1.5-2 pages.

I think the results are interesting, but they are not very complex and could be presented in a short article, maxiumum 1/3 of the current text. Select the most important results rather than presenting everything you have.

COI

All authors work for the Ugandan Medical Stores that have implemented interventions to improve MTC work.

Abstract

Rephrase first sentence: ”….irrational use of medicines may constitute 50% of xxxxxx,….” or ””of all medicine use globally, aroun 50% is considered irration….” or similar. Now it is unclear what the 50% are of.

The ”two waves” – unclear what this is. Repeated measurements?

Background:

Rephrase same sentence as in abstract.

Page 3, first para: Sentence starting with ”MTCs were identified….” is part of the methodology, not background. The following sentence: ”The aim….” should be at the end of the background section.

Length max 2 pages. No need for e.g. all the history in a research article.

Methodology:

Study design is not clear. Is this a repeated cross-sectional study with data collection conducted at two time points? When was data collection done? Who were the research team that collected data?

Statistics – this should be shortened to one paragraph – no need for formulas in a research article.

Length of methodology max 2 pages

Results:

Too long! E.g. the first paragraph is completely unnecessary. In addition to unnecessarily mentioning what sections will follow, you repeat the methodology.

Section 3.2.1 has a discussion about what statistical method was chosen – delete!

You mix in discussion in the results, e.g. page 17, first paragraph, sentence starting “this means….”. Discussion should not be put in results.

I think you need to reduce the results to maximum 1/4 of what you have here. The text is too long and difficult to follow. If you want to publish all these results, I suggest you split them into different articles. As it is now, the number of figures and results make it impossible to understand what is important and the reader is lost.

Discussion

The discussion should discuss your findings. At it reads now, it is a general overview of the status of MTCs in Uganda and elsewhere, but you have not discussed your findings in relation to this. The discussion should also be shortened but maybe only by about ½ of what you have now. However, the most important issue here is that you need to discuss your results. And the conclusion should be a summary statement of the discussion, not something new.

Reviewer #2: The authors have submitted a thesis. This is not written for a journal. The authors need to re-write the manuscript in the form of a journal articles. The introduction, methods, results, and discussion are too long, they appear to be directly lifted from a project thesis. All the sections need to be re-written and shortened.

**Do you want your identity to be public for this peer review?** For information about this choice, including consent withdrawal, please see our Privacy Policy

Reviewer #1: **Yes:** Jaran Eriksen

Reviewer #2: No

---

## [Author Response · Author response to Decision Letter 1]

22 Sep 2025

Following reviewing of the above manuscript we hereby submit a revised version after addressing the comments from the editor and the two reviewers. I have provided feedback and enriched the manuscript accordingly. Changes as guided by the editor and the two reviewers’ comments have been submitted in a separate file titled "Response to Reviewers". The changes are well reflected in the revised version of the manuscript. The entire manuscript has been significantly reduced from 49 pages to 27 pages including references.

---

## [Decision Letter · Decision Letter 1]

30 Dec 2025

Dear Dr. Nsubuga,

Thank you for submitting your manuscript to PLOS ONE. After careful consideration, we feel that it has merit but does not fully meet PLOS ONE’s publication criteria as it currently stands. Therefore, we invite you to submit a revised version of the manuscript that addresses the points raised during the review process.

We look forward to receiving your revised manuscript.

Kind regards,

Joseph Olusesan Fadare

Academic Editor

PLOS One

Journal Requirements:

Additional Editor Comments:

General comments.

The manuscript is still very voluminous and readers may get lost reading through it. The “Methods“ section can still be reduced further.

There are also too many tables and figures. Some of these may be moved to “Supplementary materials” section. It is also important to note that not all the results of the study needs to be displayed and discussed - only the relevant ones!

Is the numbering of the sections according to the journal’s format? If not, kindly revise.

Reviewers' comments:

Reviewer's Responses to Questions

**Comments to the Author**

Reviewer #2: All comments have been addressed

Reviewer #3: (No Response)

2. Is the manuscript technically sound, and do the data support the conclusions?

Reviewer #2: Yes

Reviewer #3: No

3. Has the statistical analysis been performed appropriately and rigorously?

Reviewer #2: Yes

Reviewer #3: Yes

4. Have the authors made all data underlying the findings in their manuscript fully available?

Reviewer #2: Yes

Reviewer #3: Yes

5. Is the manuscript presented in an intelligible fashion and written in standard English?

Reviewer #2: Yes

Reviewer #3: Yes

Reviewer #2: The authors have significantly improved the Manuscript. However, they should remove the numbers before the subheadings; this is not s thesis. Removing the numbers will improve the manuscript.

Page 16 mentions a Fig 4: Frequency of MTC meetings in HFs which is not shown in the manuscript, please authors should correct this or delete the mention.

Reviewer #3: (No Response)

**Do you want your identity to be public for this peer review?** For information about this choice, including consent withdrawal, please see our Privacy Policy

Reviewer #2: **Yes:** Dr. Obaro Michael

Reviewer #3: No

---

## [Author Response · Author response to Decision Letter 2]

7 Jan 2026

We hereby submit a revised version after addressing the comments from the editor and the reviewer

---

## [Editor Report · Decision Letter 2]

25 Jan 2026

Factors Influencing the Functionality of Medicines and Therapeutic Committees in Public Health Facilities in Uganda: A Longitudinal Assessment

PONE-D-25-14259R2

Dear Dr. Nsubuga,

We’re pleased to inform you that your manuscript has been judged scientifically suitable for publication and will be formally accepted for publication once it meets all outstanding technical requirements.

Kind regards,

Joseph Olusesan Fadare

Academic Editor

PLOS One

Additional Editor Comments (optional):

Thank you for addressing the issues raised during the review.
---

## [Editor Report · Acceptance letter]

PONE-D-25-14259R2

PLOS One

Dear Dr. Nsubuga,

I'm pleased to inform you that your manuscript has been deemed suitable for publication in PLOS One. Congratulations! Your manuscript is now being handed over to our production team.

Kind regards,

on behalf of

Dr. Joseph Olusesan Fadare

Academic Editor

PLOS One